# A novel method based on clustering and decision-making for construction project portfolio selection

Mohammad Khalilzadeh[1], Peyman Taebi[2], Ali Heidari[3]*

1 CENTRUM Católica Graduate Business School, Pontificia Universidad Católica del Perú, Lima, Peru,
2 School of Management and Economics, Science and Research Branch, Islamic Azad University, Tehran,
Iran, 3 School of Industrial Engineering, College of Engineering, University of Tehran, Tehran, Iran

* ali.heidari1990@ut.ac.ir

Istanbul Universitesi, TÜRKIYE

**Peer Review History:** PLOS recognizes the
benefits of transparency in the peer review
process; therefore, we enable the publication
of all of the content of peer review and
author responses alongside final, published
articles. The editorial history of this article is
available here: https://doi.org/10.1371/journal.
pone.0338697

## Abstract

Nowadays, the problem of project portfolio selection is one of the important tasks in
many construction organizations, especially project-based ones. On the other hand,
project portfolio selection usually faces many challenges due to the complexity of
project evaluation as well as limited resources. The present research aims to present
a new method based on clustering and decision-making for project portfolio selection
in project-based companies. The proposed integrated method includes the K-means
method for clustering projects, the SWARA method for prioritizing the identified crite-
ria, and the MULTIMOORA method for ranking and selecting the projects of the stud-
ied company. In addition, the results of the MULTIMOORA method was compared
with the results of the WASPAS method for verification. First, five criteria (including
18 sub-criteria) were selected using literature review and expert judgment for cluster-
ing and ranking project portfolios. Then, the research data was collected from a ques-
tionnaire containing identified criteria and sub-criteria. Based on the obtained results,
25 available construction projects were placed and ranked in 4 clusters. The findings
show that the proposed integrated method was able to cluster the project portfolios
and select the best project portfolios by ranking the project portfolios based on the
identified criteria and sub-criteria. Also, the findings indicate that the rankings using
five main criteria were different from the rankings using 18 sub-criteria, and therefore
due to the nature of the sub-criteria and the importance of paying attention to the
"desirability or undesirability of the criteria/sub-criteria in using the MULTIMOORA",
the rankings using the sub-criteria were more preferable.

## 1. Introduction

In today's rapidly evolving global business landscape, organizations face unprece-
dented challenges that stem from environmental volatility, technological disruption,
and the intensification of competition. The ability to adapt to these changes and

**Data availability statement:** The data that support the findings of this study are available within the manuscript.

**Funding:** The author(s) received no specific funding for this work.

**Competing interests:** The authors have declared that no competing interests exist.

effectively allocate scarce resources is paramount for long-term sustainability. In this context, project portfolio management (PPM) has emerged as a key mechanism for organizations to align their project initiatives with broader strategic goals while optimizing resource utilization. With the increasing complexity and risk associated with projects, organizations must carefully select a portfolio of projects that not only deliver value but also mitigate risks and avoid resource wastage. Therefore, selecting the right set of projects that align with organizational strategies, meet critical success factors, and effectively use available resources has become an essential task for organizational survival and competitiveness [1,2].

A project portfolio is often referred to as a strategic tool or "powerful strategic weapon" for implementing and achieving an organization's strategic objectives [3]. As organizations expand globally and face increased pressures from both internal and external environments, ensuring that the chosen projects align with long-term strategies has become more difficult. The process of project portfolio selection (PPS) involves evaluating a set of projects based on a variety of criteria, such as resource availability, time constraints, expected return on investment, strategic alignment, and risk factors. The complexity of this task is heightened by the fact that organizations must manage competing demands for resources, conflicting goals, and uncertainty regarding future conditions [4,5]. Poor project selection can result in wasted resources, misalignment with corporate goals, and failure to achieve strategic objectives, potentially endangering the organization's future [6,7]. This has led to a growing interest in methods that can support decision-making in the project portfolio selection process.

One of the primary challenges in the project portfolio selection process is the need to evaluate multiple competing factors simultaneously, especially in a multi-project environment where resources are limited. Organizations often encounter a range of complexities, such as resource constraints, labor shortages, tight timelines, conflicting stakeholder interests, and various forms of uncertainty, all of which complicate the decision-making process [8]. Furthermore, strategic goals are often multifaceted, and the process of selecting the optimal set of projects must reflect both short-term and long-term objectives. The critical importance of selecting the right portfolio has resulted in the development of various methodologies to guide decision-makers in evaluating project candidates, balancing risks, and ensuring alignment with organizational strategies [9,10].

Recent advancements in Multi-Criteria Decision-Making (MCDM) techniques have significantly contributed to the field of project portfolio selection. MCDM methods, such as the MULTIMOORA framework, have gained attention due to their ability to evaluate and rank projects based on multiple criteria, thus allowing decision-makers to capture the complexity inherent in real-world project portfolio decisions [11]. These methods offer a systematic approach to handle both qualitative and quantitative criteria and support more informed, data-driven decisions. The need for comprehensive MCDM frameworks is particularly pressing as organizations now face a greater diversity of projects with varying goals, risks, and strategic value. The ability to integrate decision-makers' preferences and incorporate uncertainty in MCDM models has opened new avenues for improving the accuracy and reliability of project portfolio decisions [12,13].

The application of clustering techniques, such as K-Means, has also emerged as a useful tool in project portfolio management, especially for categorizing projects based on shared characteristics. This approach allows for a more nuanced understanding of project types, which is critical when dealing with portfolios that span multiple domains or industries. K-Means clustering can assist in identifying similar projects, which can then be evaluated together to streamline the decision-making process. By using clustering techniques, organizations can group projects according to strategic alignment, risk profiles, and other relevant factors, making the selection process more efficient and effective [14,15].

While existing literature has made significant strides in addressing the complexities of project portfolio selection, there remains a need for a more integrated approach that combines clustering techniques with advanced MCDM methods. Recent studies have demonstrated that while many decision-making models exist, few effectively address the full range of challenges organizations face when selecting projects. There is a clear gap in methodologies that can handle multiple objectives while accounting for both qualitative and quantitative data and accommodating the ever-changing nature of project environments [16,10].

To address this gap, the present research proposes a novel approach that combines K-Means clustering with SWARA and MULTIMOORA techniques to support project portfolio selection. First, the K-Means method is applied to categorize projects based on their characteristics, allowing for a clearer understanding of project groupings and strategic relevance. Subsequently, the SWARA and MULTIMOORA methods are utilized to rank and select the most suitable projects for inclusion in the portfolio, based on multiple criteria that align with organizational goals. The SWARA and MULTIMOORA methodologies are employed for project portfolio selection due to their complementary strengths in handling complex, multi-criteria decision-making problems. SWARA is ideal for determining the relative importance of criteria based on expert judgment, making it suitable for projects where subjective assessments, such as strategic alignment and risk, are crucial [17,18]. MULTIMOORA, on the other hand, excels in evaluating alternatives based on both quantitative and qualitative criteria, offering a holistic approach to ranking and selecting projects [19,20,21,22]. Together, these methods provide a comprehensive framework that addresses both the subjective and objective aspects of project portfolio selection, ensuring alignment with organizational goals while effectively managing resources and risks.

By integrating these advanced methods, this research aims to provide a more comprehensive and adaptable framework for project portfolio selection, particularly in environments characterized by resource constraints, uncertainty, and conflicting objectives.

The structure of this paper is as follows: Section 2 presents a review of relevant studies and highlights recent advancements in project portfolio selection methodologies. Section 3 details the research methodology and the approach taken in this study. Section 4 provides the results of the proposed framework, and Section 5 concludes with a discussion of the findings, implications for practice, and recommendations for future research.

## 2. Literature review

### 2.1. Project portfolio management and selection complexity

Project portfolio selection is a critical process for organizations to ensure that their projects align with strategic goals, are feasible within resource constraints, and have the potential for high returns. Each project in a portfolio has unique characteristics, such as environmental conditions, resource requirements, and strategic objectives, making the process of project portfolio management (PPM) inherently complex [23]. According to [24], project portfolios are collections of projects with common features and goals, managed collectively to maximize organizational value. Project portfolio management includes project selection, prioritization, resource allocation, and risk assessment, all of which require a systematic approach to evaluate the various aspects of potential projects [25,26].

Effective portfolio selection is crucial to achieving organizational objectives. Failure to select appropriate projects can lead to inefficiencies, resource misallocation, and strategic misalignment [27]. According to [28], improper project selection decreases organizational credibility, increases costs, and ultimately harms profitability. Tavakoli et al. (2023) argue that

selecting projects based on clearly defined criteria and aligning them with organizational goals helps prevent such out-comes. Additionally, decision-makers must ensure that the criteria used for selection are aligned with the strategic direction of the organization [29]. The complexity of this task demands a robust, comprehensive methodology for evaluating and selecting the right mix of projects.

## 2.2. Criteria for project portfolio selection

The criteria used for project selection vary widely depending on the goals, resources, and risk tolerance of the organization. As illustrated in Table 1, the criteria can be broadly categorized into financial, organizational, technical, risk, and service quality factors [30, Pramanik et al., 2020]. These criteria allow decision-makers to assess various aspects of each project, such as required budget, return on investment, alignment with organizational strategy, access to required technology, and the potential risks involved. Recent studies, such as those by [38,39,33], have explored these criteria in detail, particularly focusing on the importance of aligning projects with organizational goals to maximize strategic benefits.

However, despite the identification of key criteria, one major challenge in project portfolio selection is that decision-makers often face projects with vastly different characteristics. This makes it difficult to directly compare projects based solely on predefined criteria. As emphasized by [25,23], project clustering has the potential to enhance portfolio selection by grouping similar projects together, making it easier to compare and evaluate them against common benchmarks. Yet, this critical aspect of clustering has often been underexplored in previous research.

**Table 1. Project portfolio selection criteria based on previous studies.**

| Category | Criterion | Source |
|---|---|---|
| **Financial** | Required budget | [30,31,32,33], Pramanik et al (2020) |
| | Project profitability | Pramanik et al (2020), [34]. |
| | Net present value | [35,34,36]. |
| | Rate of return on investment | [37,30,31,35], Pramanik et al (2020) |
| **Organizational** | Access to skilled labor | [38,39,33,31] |
| | Alignment with the goals and strategies of the organization | Demircan keskin (2019), Martinsou & Killen (2014), [40,34]. |
| | Helping to gain more market share | [41,42,31] |
| | Project location | [43,32] |
| | Experience of similar projects | [43] |
| | Support of senior managers | [44] |
| **Technical and practical** | Access to the required technology and equipment | [31] |
| | Appropriate project scheduling | [44,35,32,34,45] |
| | Advanced equipment and technology | [35] |
| | The publicity of the required technology | Hashemzade & Ju (2019), [46] |
| **Risks** | Time risks | [43,44] |
| | Cost risks | [31,35] |
| | Technical and quality risks | [31,35,34]. |
| | Safety risks | [31] |
| | Environmental and political risks | [43,35,34]. |
| **Service quality** | Improving service quality | Hashemzade & Ju (2019), [34], |
| | Increasing the decision-making power of managers | [31] |

Table 1 presents the frequent criteria that previous researchers have used in five financial, organizational, technical and practical, risks, and service quality categories:

## 2.3. Integration of MCDM methods in project portfolio selection

A variety of MCDM methods have been proposed to address the complexities of project portfolio selection. Traditional MCDM techniques, such as Analytic Hierarchy Process (AHP), TOPSIS, and VIKOR, are widely used in decision-making processes due to their ability to incorporate both qualitative and quantitative criteria [33,47]. Recent studies have applied these methods to project selection problems in various industries, including manufacturing [4], construction [36], and healthcare [Zhang et al., 2021], demonstrating their versatility and robustness. For instance, [4] used fuzzy TOPSIS to select a project portfolio based on sustainability criteria in a paper manufacturing company, showcasing the growing importance of sustainability in portfolio decisions. Similarly, [48] combined TOPSIS with the Ordinal Priority Approach (OPA) and K-means clustering to select projects for a refinery equipment manufacturing company, emphasizing the utility of clustering in improving the efficiency of project selection. This study highlighted the advantage of grouping similar projects, thus reducing the complexity and increasing the reliability of the final project selection [48]. These findings suggest that combining clustering methods with MCDM techniques can enhance the accuracy and speed of project portfolio selection by making it easier to identify projects that align with organizational priorities.

## 2.4. Recent advances in Hybrid MCDM methods

In recent years, there has been a growing trend toward hybrid MCDM approaches that combine multiple decision-making methods to address the complexity of project portfolio selection. For example, [33] proposed a fuzzy AHP-based method to select sustainable projects, while [49] employed fuzzy AHP to rank sustainable development projects according to nine criteria. These approaches focus on incorporating fuzzy logic to handle uncertainty and provide a more robust evaluation of project portfolios.

Additionally, several studies have explored the integration of clustering techniques with MCDM methods to further improve project selection. El Bok & Berrado (2022) applied clustering to categorize projects before applying MCDM methods for ranking, and [50] used the fuzzy BWM method to identify criteria and DEA for evaluating project performance. This integration allows organizations to first classify projects based on key characteristics before applying the MCDM methods to rank and select them, leading to more efficient and reliable decision-making.

## 2.5. Addressing gaps and enhancing the methodology

Despite the progress made in this area, a significant gap remains in the literature regarding the integration of clustering techniques and advanced MCDM methods, such as SWARA and MULTIMOORA, for project portfolio selection. While most studies have focused on selecting the best projects from similar ones, project-based organizations often face portfolios that include projects with highly diverse characteristics. The ability to classify these projects based on their features before selecting the most suitable ones can significantly improve the decision-making process. This study aims to fill this gap by combining K-means clustering with the SWARA and MULTIMOORA methods to provide a more comprehensive and adaptable approach to project portfolio selection. The SWARA method helps to determine the relative importance of selection criteria, ensuring that the most relevant factors are emphasized. Meanwhile, MULTIMOORA incorporates both cardinal and ordinal criteria, offering a multi-dimensional evaluation framework that allows for more nuanced comparisons of project alternatives [36,51]. The dual-method approach allows for more robust decision-making by addressing both subjective preferences (via expert judgment in SWARA) and objective performance measures (via MULTIMOORA). This hybrid methodology offers a more reliable ranking of projects and provides a systematic process that can be adapted to the specific needs of different organizations, industries, and project portfolios.

This literature review has highlighted the critical role of project portfolio selection and the use of advanced MCDM methods, particularly when dealing with diverse and complex project portfolios. The integration of K-means clustering with SWARA and MULTIMOORA offers a significant contribution to the literature by combining clustering techniques with multi-criteria decision analysis to improve the selection process. This hybrid approach not only enhances the reliability and accuracy of project ranking but also provides a flexible, adaptable framework that can be tailored to the specific needs of different industries. By addressing the gap in the existing research on project clustering and selection, this study contributes to advancing both academic understanding and practical applications of project portfolio management.

While existing MCDM-based models have addressed various facets of project portfolio selection, the main contribution of the present study lies in the integrated use of K-means clustering, SWARA, and MULTIMOORA, a combination not previously applied in a unified framework for construction project portfolio selection. Unlike prior studies, which either rely on single MCDM approaches or basic clustering integration, the proposed model first categorizes heterogeneous projects via unsupervised learning (K-means), then assigns criteria weights using expert-driven SWARA, and finally applies the robust MULTIMOORA technique for ranking, enabling a more granular and adaptive decision-making process. This integrated hybrid model addresses the often-overlooked challenge of project diversity within portfolios and provides a structured approach to improve decision accuracy, reliability, and contextual adaptability.

In summary, the present study contributes to the literature by introducing a novel, integrated methodology for project portfolio selection that combines clustering with advanced MULTIMOORA and SWARA methods. This approach not only enhances decision-making in practice but also offers new avenues for future research in both project management and multi-criteria decision analysis.

## 3. Methodology

The current research was conducted with the aim of providing a new method for selecting proper projects in Pars Garma Construction & Industrial Company which operates in multiple civil engineering sectors, including dam and dike construction, irrigation and drainage execution, road and bridge construction, drilling and tunnel work, heavy concrete and metal constructions, social housing development, and the design and construction of production plants. Considering the wide range of this organization's activities in numerous fields and the existence of projects with different characteristics, the projects were first classified based on their characteristics and then the projects in each category were ranked according to the desirable criteria in order to select the right portfolio of projects.

The proposed methodology for selecting the project portfolio is presented in Fig 1. According to the process displayed in Fig 1, first the criteria presented in Table 1 were screened and localized. Next, the projects were clustered using the K-means algorithm, then the weight of each criterion was determined using the SWARA method. Subsequently, the projects in each category were ranked using the MULTIMOORA method according to the desired criteria, and eventually the results were compared with the results of the WASPAS method for verification.

The decision-making panel in this research consisted of 23 senior managers and project managers of the company selected based on three key criteria: (1) minimum 10 years of industry experience, (2) direct involvement in project portfolio decisions, and (3) representation across all operational domains (finance, technical, risk management, and strategic planning), and the required data was collected using the three following questionnaires: (1) Questionnaire for screening the project selection criteria, (2) Questionnaire for weighting the criteria using the SWARA method, and (3) Questionnaire for comparing the projects using the MULTIMOORA method based on the criteria. Fig 1 illustrates the steps of the research methodology.

### 3.1. Clustering with K-MEANS algorithm

The K-means algorithm is faster than other hierarchical clustering algorithms and is used for a large amount of data and tries to estimate the following for a fixed number of clusters:

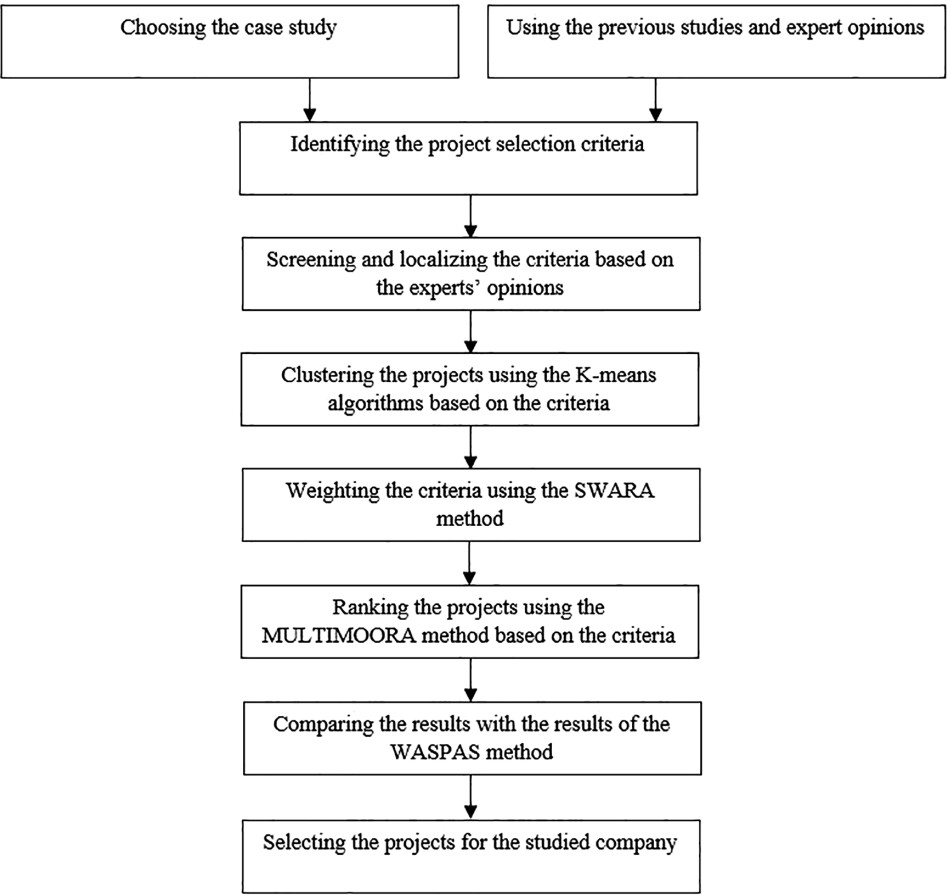

**Fig 1. The steps of the research methodology.**

(1) Obtaining points as the centers of clusters, which are actually the average points belonging to each cluster,

(2) Attributing each data sample to a cluster that has the smallest distance to the center of that cluster.

In the K-means algorithm, the number of clusters is not known in advance and the clusters do not have a common interface, hence, Equation (1) is used to create the clusters:

$$J(u, v) = \sum_{j=1}^{k} \sum_{i=1}^{n} d_{ij}^2$$

(1)

where $J(u,v)$ is the fitness function (objective), $d$ is the Euclidean distance, $n$ is the number of data samples, and $k$ is the number of cluster centers. This algorithm starts with an initial random division and tries to attribute each sample to the cluster that the sample has the most similarity with its central elements. Then, the new central elements of each cluster are calculated and replace the previous central elements. This operation continues until reaching a convergence criterion. In the K-means algorithm, convergence is achieved when a sample from one cluster cannot be assigned to another cluster or when there is no change in the centers of the clusters, and the algorithm stops in this situation. Different initial values for the K-means algorithm can lead to different clustering. Since this algorithm is based on Euclidean distance, it

 

can converge to the local minimum. This is usually true for clusters that are not very well separated. It has been shown that there is no guarantee for the convergence of an iterative algorithm to a global optimum [52].

The best clustering is to maximize the total similarity between the cluster center and all cluster members and minimize the total similarity between the cluster centers. In order to choose the best cluster, a suggested range for clusters is determined based on expert opinions and previous studies. Then, the value of $\rho(k)$ is calculated for each of the $k$ values. The value of $k$ at which $\rho(k)$ is maximized is chosen as the optimal number of clusters. In this way, it is possible to choose the number of clusters for which the distance between the cluster centers and the similarity of the cluster centers with the members within each cluster is maximum. The quality of clustering results with $k$ clusters is defined as follows:

$$o = \left\{ C^n \middle| n = 1, \cdots, k \right\} \tag{2}$$

$$O^n = \left\{ C_i \middle| i = 1, \cdots, \| T^c - o \| \right\} \tag{3}$$

$$\rho(k) = \frac{1}{k} \sum_{n=1}^{k} min \left\{ \frac{\eta_n + \eta_m}{\delta_{mn}} \right\} \tag{4}$$

$$\eta_n = \frac{1}{\|O^n\|} \sum_{C_i \in O^n} sim(C_i, O^n) \tag{5}$$

$$\eta_m = \frac{1}{\|O^m\|} \sum_{C_i \in O^m} sim(C_i, O^m) \tag{6}$$

$$\delta_{nm} = sim(C^n, C^m) \tag{7}$$

In the above equations, $O$ is the set of cluster centers, $C^n$ is the cluster centers, $O^n$ is the set of elements that are not selected as cluster centers, $T^c$ is the set of all elements on which clustering has been done, $\eta^n$ is the average similarity between the cluster centers, $C^n$, and all the elements of the cluster $O^n$, $\eta^m$ is the average similarity between the center of the cluster $C^m$ and all the elements of the cluster $O^m$, and $\delta^{mn}$ is the similarity of $C^n$ and $O^n$ [52].

### 3.2. The SWARA method

The SWARA method has a simple mathematical structure and allows experts to choose their desired priorities based on the current conditions. In other words, in this method, no pairwise comparisons are performed, so it does not face the compatibility issues of pairwise comparisons, which can have a smaller computational volume compared to the other weighting methods. SWARA has the ability to evaluate the opinion of experts about the ratio of importance of criteria in determining their weight. This method gives the decision maker the opportunity to choose their priorities based on the current situation of the organization and its environment, which means that the decision maker ranks the criteria according to the current conditions The steps of this method are as follows [53]:

**Step 1:** First, experts vote for their desired criteria, and then the percentage of votes for each index is calculated by dividing the number of votes for that index by the number of experts. The percentage of votes is the relative frequency of each index. In this step, the indicators are ranked according to the percentage of votes and in descending order.

**Step 2:** The relative difference of votes of each criterion compared to the previous criterion *(Sj)* is calculated. This value is not calculated for the criterion with the highest number of votes.

**Step 3:** The growth parameter *(Kj)* which is equal to 1 for the criterion with the highest rating (the criterion with the first rank or the most important criterion) and for other criteria is calculated from Equation (8):

$$K_j = S_j + 1, \ j \neq 1$$

(8)

**Step 4:** The value of 1 is considered for *qj* corresponding with the most important criterion and is calculated by using Equation (9) for other criteria:

$$q_j = \frac{q_{j-1}}{K_j} \ , j \neq 1$$

(9)

**Step 5:** Finally, the weight of each index is obtained through Equation (10):

$$W_j = \frac{q_j}{\sum_{j=1}^{n} q_j}$$

(10)

### 3.3. The MULTIMOORA method

This method was presented in 2010 by [54]. The steps of this method are as follows:

**Step 1:** Formation of the decision matrix:

$$X = [x_{ij}]_{n \times m} = \begin{bmatrix} x_{11} x_{12} \dots x_{1m} \\ x_{21} x_{22} \dots x_{2m} \\ \dots \\ x_{n1} x_{n2} \dots x_{mm} \end{bmatrix}$$

(11)

In Equation (11), *X* is the decision matrix, *i* are the alternatives and j are the criteria of the problem.

**Step 2:** Normalization of the decision matrix:

$$n_{ij} = \frac{x_{ij}}{\sqrt{\sum_{i=1}^{m} x_{ij}^2}}$$

(12)

where $n_{ij}$ is the normalized value of $x_{ij}$ in the decision matrix.

**Step 3:** Ranking the alternatives based on ratio system:

$$y_j^* = \sum_{i=1}^{i=g} W_j n_{ij} - \sum_{i=g+1}^{i=n} w_j n_{ij}$$

(13)

In Equation (13), $y_j^*$ is the score of each option based on the ratio system, $w_j$ is the weight of each criterion, *g* are indicators with a positive nature and *g + 1* are indicators with a negative nature.

**Step 4:** Ranking the alternatives based on the reference point approach: In this step, the reference point must first be obtained for each criterion in the normalized matrix. The reference point for criteria with a positive nature is equal to the largest value of options based on that criterion and for negative criteria it is equal to the smallest value of options based on that criterion in the normalized decision matrix. We have:

For positive criteria:

$$r_j = \max_j w_j n_j \qquad (14)$$

For negative criteria:

$$r_j = \min_j w_j n_j \qquad (15)$$

The ranking of options in the reference point approach is obtained from Equation (16):

$$\min_j \left\{ \max_i \left| w_j r_i - w_j n_{ij} \right| \right\} \qquad (16)$$

**Step 5:** Obtaining the completely multiplicative index and ranking the alternatives:

$$U_i = \frac{\prod_{j=1}^{g} \left( w_j x_{ij}^* \right)}{\prod_{j=g+1}^{n} \left( w_j x_{ij}^* \right)} \qquad (17)$$

**Step 6:** MULTIMOORA final ranking: This method provides three approaches with three ratings for alternatives, and finally, the dominance theory is used to integrate these ratings. In other words, the final ranking is derived using the dominance theory, which aggregates the rankings from the ratio system, reference point, and full multiplicative approaches. The principle prioritizes alternatives that consistently perform well across all three methods, resolving ties by hierarchical precedence [55,56].

### 3.4. The WASPAS method

The WASPAS method was introduced by [57] and is a combination of the weighted sum method (WSM) and the weighted product method (WPM). The accuracy of this method is higher than other independent methods. The steps of this method are as follows [19]:

   **Step 1:** The decision matrix, denoted by $r_{ij}$, is transformed into a linearly dimensionless form.
   **Step 2:** Calculate the multiplicative nondimensional matrix using Equation (18):

$$t_{ij}^{(1)} = r_{ij} \times w_j \qquad (18)$$

where $w_j$ is the weight of the criterion.
   **Step 3:** Calculate the summation of the rows of the multiplicative nondimensional matrix using Equation (19):

$$U_i^{(1)} = \sum_{j=1}^{n} t_{ij}^{(1)} \qquad (19)$$

**Step 4:** Calculate the power-weighted nondimensional matrix using Equation (20):

$$t_{ij}^{(2)} = r_{ij}^{(w_j)} \qquad (20)$$

**Step 5:** Calculate the row product of the power-weighted nondimensional matrix using Equation (21):

$$U_i^{(2)} = \prod_{j=1}^{n} t_{ij}^{(2)}$$

(21)

**Step 6:** Calculate the average score of the multiplicative and power-weighted nondimensional matrices using Equation (22):

$$U_i = 0.5 \left[ U_i^{(1)} + U_i^{(2)} \right]$$

(22)

**Step 7:** Rank the options in descending order based on $U_i$.

## 4. Results

There were 25 projects under implementation in the studied company and all those 25 projects were examined in this research.

### 4.1. Screening research criteria

According to the review of previous studies, 5 criteria and 18 sub-criteria that have the most frequency and importance were identified as presented in Table 2. In order to screen and localize these criteria and determine the most important and practical ones, a questionnaire based on the 7-point Likert scale was designed and provided to the decision-making panel. These ranges are: "very important (7)", "important (6)", "slightly important (5)", "neutral (4)", "slightly unimportant (3)", "unimportant (2)", "very unimportant (1)" [58]. Then, according to the opinions of the decision-making panel members, the criteria that had an average above 4 were taken into consideration as clustering criteria in the next steps.

**Table 2. The final research sub-criteria.**

| Row | Code | Criteria | Type | Sub-criteria |
|-----|------|----------|------|--------------|
| 1 | SC1 | **Financial** | Undesirable | Required budget |
| 2 | SC2 | | Desirable | Project profitability |
| 3 | SC3 | | Undesirable | Net present value |
| 4 | SC4 | **Technical and practical** | Desirable | Access to the required technology and equipment |
| 5 | SC5 | | Desirable | Appropriate project scheduling |
| 6 | SC6 | | Desirable | Advanced equipment and technology |
| 7 | C7 | **Service quality** | Desirable | Improving service quality |
| 8 | SC8 | | Desirable | Increasing the decision-making power of managers |
| 9 | SC9 | **Organizational** | Desirable | Access to skilled labor |
| 10 | SC10 | | Desirable | Alignment with the goals and strategies of the organization |
| 11 | SC11 | | Desirable | Helping to gain more market share |
| 12 | SC12 | | Desirable | Experience of similar projects |
| 13 | SC13 | | Desirable | Support of senior managers |
| 14 | SC14 | **Risks** | Undesirable | Time risks |
| 15 | SC15 | | Undesirable | Cost risks |
| 16 | SC16 | | Undesirable | Technical and quality risks |
| 17 | SC17 | | Undesirable | Safety risks |
| 18 | SC18 | | Undesirable | Environmental and political risks |

## 4.2. Determining the weights of the criteria

The SWARA method was used to determine the weights of the criteria. For this purpose, the members of the decision-making panel were asked to specify the important criteria in their opinions, the results of which are provided in Table 3.

Finally, the obtained weights are given in Table 4.

## 4.3. Clustering of projects with the K-means method

In this research, the K-means method and SPSS version 25 software were used to cluster the projects. For this purpose, first, the opinions of the members of the decision-making panel were collected for 5 criteria and 18 identified sub-criteria using a researcher-made questionnaire. Then, the collected data were analyzed using the K-means method in SPSS software. The findings showed that 25 available projects were classified in 4 clusters (regarding 5 criteria and 18 sub-criteria). Considering the existence of 5 criteria to evaluate projects, the results of the clustering algorithm were significant only for 4 clusters. Also, the number of optimal clusters was checked by R-STUDIO software, the output of which is depicted in Fig 2.

Table 5 exhibits the values related to the centers of each cluster according to the five main criteria used for clustering.

Table 6 shows the values related to the centers of each cluster regarding 18 criteria for clustering.

Table 7 presents the amount of changes in the center of each cluster in repeating the clustering algorithm for all four clusters using five criteria.

Table 8 indicates the changes of cluster centers in the iteration of the clustering algorithm considering 18 sub-criteria which confirms that the clustering was done in the first iteration.

Table 9 shows the final center of each of the four clusters based on the five criteria for clustering. This table is the final table and is presented after the finalization of the calculations, so its values are different from the initial values of the cluster centers (shown in Table 5).

Table 10 exhibits the final values of the cluster centers separately, based on which the final values of the cluster centers are different from the initial values of the cluster centers (shown in Table 6).

The most important output of K-means is the ANOVA table. The ANOVA results show that each criterion or sub-criterion is used in one cluster or more than one cluster. For the criterion to be used in a cluster, it is necessary to have a significance level less than or equal to 0.05. Otherwise, the criteria are used in more than one cluster for clustering. Of course, this does not affect the accuracy of clustering. Rather, it is only an explanation of the situation of using the criteria in the clustering process. Table 11 presents the results of the ANOVA test for clustering according to five criteria.

**Table 3. The number of votes given to the sub-criteria by decision-makers.**

| Criterion | SC1 | SC2 | SC3 | SC4 | SC5 | SC6 | SC7 | SC8 | SC9 |
|---|---|---|---|---|---|---|---|---|---|
| Number of votes | 22 | 20 | 12 | 13 | 18 | 12 | 20 | 18 | 21 |
| Criterion | SC10 | SC11 | SC12 | SC13 | SC14 | SC15 | SC16 | SC17 | SC18 |
| Number of votes | 15 | 17 | 17 | 19 | 11 | 18 | 17 | 15 | 13 |

**Table 4. The final weights of the sub-criteria.**

| Criterion | SC1 | SC2 | SC3 | SC4 | SC5 | SC6 | SC7 | SC8 | SC9 |
|---|---|---|---|---|---|---|---|---|---|
| Weight | 0.064 | 0.059 | 0.048 | 0.063 | 0.054 | 0.056 | 0.056 | 0.058 | 0.051 |
| Criterion | SC10 | SC11 | SC12 | SC13 | SC14 | SC15 | SC16 | SC17 | SC18 |
| Weight | 0.057 | 0.048 | 0.061 | 0.057 | 0.047 | 0.057 | 0.056 | 0.054 | 0.051 |

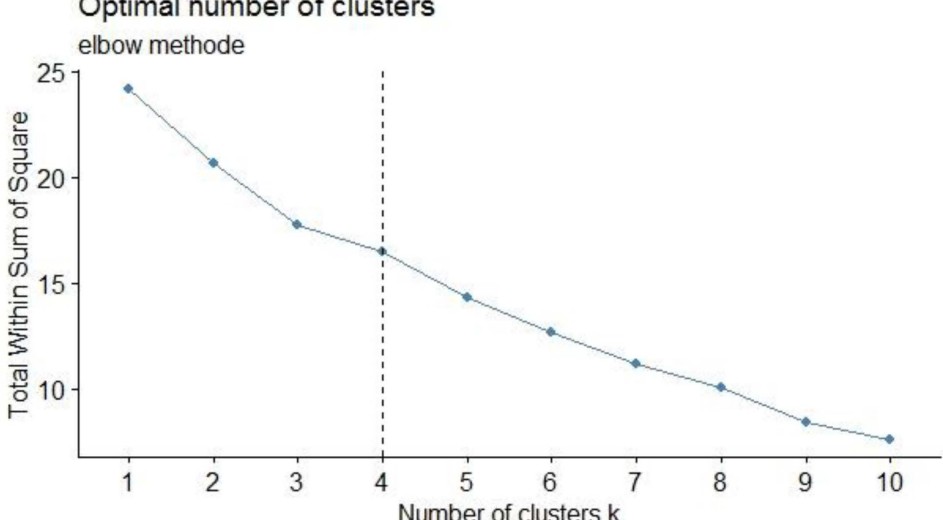

**Fig 2. Optimal number of clusters.**

**Table 5. The values of cluster centers based on five main criteria.**

| Criteria | First cluster | Second cluster | Third cluster | Forth cluster |
|---|---|---|---|---|
| Financial | 5.81 | 5.21 | 5.38 | 5.67 |
| Technical and practical | 4.92 | 4.65 | 5.19 | 5.75 |
| Service quality | 5.84 | 5.38 | 5.09 | 5.47 |
| Organizational | 5.68 | 5.53 | 5.59 | 5.79 |
| Risks | 5.43 | 5.28 | 5.49 | 5.58 |

Also, Table 12 shows the results of the ANOVA test for clustering regarding 18 sub-criteria.

Finally, Table 13 displays the number of projects in each cluster. As can be seen in this table, the clustering results based on five main criteria differ significantly from those based on 18 sub-criteria. While it is not possible to determine which method is more accurate, researchers prefer the results from sub-criteria clustering for their greater precision.

Then, the findings of the ranking of projects in each cluster are presented and analyzed using the MULTIMOORA method.

## 4.4. Ranking of projects in each cluster using the MULTIMOORA method

After clustering 25 available projects, the MULTIMOORA method was exploited to rank them. For this purpose, among the 18 sub-criteria presented in Table 2, 7 sub-criteria were considered as unfavorable criteria and the rest as favorable sub-criteria. Table 14 presents the values of the initial decision matrix for all 25 projects according to 18 sub-criteria.

Next, after the initial decision matrix was normalized and weighted based on the weights of the sub-criteria, resulting from the SWARA method, the rankings of the projects are presented in Table 15.

Therefore, the projects in the studied company were clustered and ranked by using the proposed methodology according to the 5 criteria and 18 sub-criteria. Finally, Table 16 shows the rankings of the projects in each cluster based on 18 sub-criteria.

**Table 6. The values of cluster centers according to 18 criteria.**

| Sub-criteria | First cluster | Second cluster | Third cluster | Forth cluster |
|---|---|---|---|---|
| Required budget | 6.56 | 5.63 | 5.94 | 6.13 |
| Project profitability | 5.75 | 5.44 | 5.25 | 5.56 |
| Net present value | 5.13 | 4.56 | 4.88 | 5.31 |
| Access to the required technology and equipment | 4.81 | 4.88 | 5.25 | 5.94 |
| Appropriate project scheduling | 5.31 | 4.75 | 5.38 | 6.13 |
| Advanced equipment and technology | 4.63 | 4.31 | 5.06 | 5.19 |
| Improving service quality | 5.81 | 5.31 | 5.69 | 5.38 |
| Increasing the decision-making power of managers | 5.88 | 5.44 | 5.44 | 5.56 |
| Access to skilled labor | 5.81 | 5.75 | 6.13 | 5.81 |
| Alignment with the goals and strategies of the organization | 5.44 | 5.38 | 5.88 | 5.69 |
| Helping to gain more market share | 5.38 | 5.19 | 5.19 | 6.06 |
| Experience of similar projects | 5.81 | 5.88 | 5.50 | 5.88 |
| Support of senior managers | 5.94 | 5.44 | 5.25 | 5.50 |
| Time risks | 4.81 | 4.75 | 5.38 | 5.44 |
| Cost risks | 5.69 | 5.38 | 5.06 | 5.69 |
| Technical and quality risks | 5.44 | 5.19 | 5.44 | 5.75 |
| Safety risks | 5.69 | 5.88 | 5.94 | 5.75 |
| Environmental and political risks | 5.50 | 5.19 | 5.31 | 5.25 |

**Table 7. Changes of cluster centers in iterations of the clustering algorithm.**

| Iteration | First cluster | Second cluster | Third cluster | Forth cluster |
|---|---|---|---|---|
| 1 | 0.243 | 0.264 | 0.285 | 0.187 |
| 2 | 0.087 | 0.107 | 0.025 | 0.000 |
| 3 | 0.000 | 0.000 | 0.000 | 0.000 |

**Table 8. Changes of cluster centers in iterations of the clustering algorithm.**

| Iteration | First cluster | Second cluster | Third cluster | Forth cluster |
|---|---|---|---|---|
| 1 | 0.855 | 0.798 | 0.846 | 0.553 |
| 2 | 0.000 | 0.000 | 0.000 | 0.000 |

**Table 9. Final cluster centers of five criteria.**

| Criteria | First cluster | Second cluster | Third cluster | Forth cluster |
|---|---|---|---|---|
| Financial | 5.74 | 5.28 | 5.49 | 5.61 |
| Technical and practical | 4.99 | 4.90 | 5.13 | 5.61 |
| Service quality | 5.74 | 5.53 | 5.32 | 5.52 |
| Organizational | 5.65 | 5.59 | 5.57 | 5.70 |
| Risks | 5.30 | 5.41 | 5.38 | 5.51 |

According to Table 15, the results of the MULTIMOORA and WASPAS methods are very close to each other. In some projects, the results are completely consistent with each other, and in others, the results are consistent with one of the three methods in the MULTIMOORA technique. Therefore, the two methods confirm each other's results to some extent.

**Table 10. Final values of the cluster centers.**

| Sub-criteria | First cluster | Second cluster | Third cluster | Forth cluster |
|---|---|---|---|---|
| Required budget | 6.17 | 5.72 | 5.88 | 6.19 |
| Project profitability | 5.69 | 5.46 | 5.50 | 5.53 |
| Net present value | 5.11 | 4.76 | 5.10 | 5.13 |
| Access to the required technology and equipment | 5.14 | 5.04 | 5.13 | 5.84 |
| Appropriate project scheduling | 5.20 | 5.05 | 5.24 | 5.97 |
| Advanced equipment and technology | 4.88 | 4.68 | 5.00 | 5.03 |
| Improving service quality | 5.52 | 5.32 | 5.27 | 5.53 |
| Increasing the decision-making power of managers | 5.66 | 5.56 | 5.44 | 5.50 |
| Access to skilled labor | 5.59 | 5.59 | 5.78 | 5.94 |
| Alignment with the goals and strategies of the organization | 5.61 | 5.44 | 5.75 | 5.50 |
| Helping to gain more market share | 5.30 | 5.48 | 5.48 | 5.88 |
| Experience of similar projects | 5.59 | 5.75 | 5.58 | 5.72 |
| Support of senior managers | 5.75 | 5.56 | 5.39 | 5.47 |
| Time risks | 5.91 | 5.06 | 5.23 | 5.47 |
| Cost risks | 5.61 | 5.55 | 5.31 | 5.63 |
| Technical and quality risks | 5.20 | 5.34 | 5.39 | 5.50 |
| Safety risks | 5.70 | 5.77 | 5.78 | 5.72 |
| Environmental and political risks | 5.36 | 5.29 | 5.12 | 5.25 |

**Table 11. The ANOVA Results of clustering based on five criteria.**

| | Cluster | | Error | | F | Meaningfulness | Clustering status with criteria |
|---|---|---|---|---|---|---|---|
| | Mean square | df | Mean square | df | | | |
| **Criterion 1** | 0.148 | 3 | 0.008 | 21 | 17.717 | 0.000 | Used in one cluster |
| **Criterion 2** | 0.257 | 3 | 0.018 | 21 | 14.604 | 0.000 | Used in one cluster |
| **Criterion 3** | 0.175 | 3 | 0.019 | 21 | 9.310 | 0.000 | Used in one cluster |
| **Criterion 4** | 0.113 | 3 | 0.010 | 21 | 1.390 | 0.274 | Used in more than one cluster |
| **Criterion 5** | 0.118 | 3 | 0.011 | 21 | 1.670 | 0.204 | Used in more than one cluster |

Since the MULTIMOORA method uses three methods of the ratio system, the reference point, and the full multiplicative ranking simultaneously for the final ranking, it is obvious that the use of three methods is stronger than two methods or one method. Therefore, the results of the MULTIMOORA technique are more valid [54] and the use of the MULTIMOORA method is recommended.

Fig 3 displays as a comparison of the rankings obtained by MULTIMOORA and WASPAS. The closeness of the rankings obtained by the two methods can be seen in this figure.

## 4.5. Sensitivity analysis

In order to analyze the sensitivity of the identified criteria, the projects were ranked in the condition of removing each of the main criteria and the results were compared with the original ranking (considering all 5 main criteria). The results are presented in Table 17.

According to the results, the least change in the ranking occurs by removing the sub-criteria of the service quality. Considering that according to the SWARA method, this main criterion has acquired the least importance, the results of the sensitivity analysis confirm the results of the SWARA method.

**Table 12. The ANOVA Results of clustering based on 18 sub-criteria.**

|  | Cluster | | Error | | F | Meaningfulness | Clustering status with criteria |
|---|---|---|---|---|---|---|---|
|  | Mean square | df | Mean square | df |  |  |  |
| **SC1** | 0.220 | 3 | 0.029 | 21 | 7.519 | 0.001 | Used in one cluster |
| **SC2** | 0.046 | 3 | 0.054 | 21 | 0.848 | 0.483 | Used in more than one cluster |
| **SC3** | 0.178 | 3 | 0.039 | 21 | 4.537 | 0.013 | Used in one cluster |
| **SC4** | 0.344 | 3 | 0.036 | 21 | 9.548 | 0.000 | Used in one cluster |
| **SC5** | 0.426 | 3 | 0.032 | 21 | 13.333 | 0.000 | Used in one cluster |
| **SC6** | 0.151 | 3 | 0.043 | 21 | 3.493 | 0.034 | Used in one cluster |
| **SC7** | 0.085 | 3 | 0.058 | 21 | 1.455 | 0.255 | Used in more than one cluster |
| **SC8** | 0.054 | 3 | 0.038 | 21 | 1.436 | 0.260 | Used in more than one cluster |
| **SC9** | 0.102 | 3 | 0.071 | 21 | 1.440 | 0.260 | Used in more than one cluster |
| **SC10** | 0.143 | 3 | 0.030 | 21 | 4.830 | 0.010 | Used in one cluster |
| **SC11** | 0.149 | 3 | 0.056 | 21 | 2.674 | 0.074 | Used in more than one cluster |
| **SC12** | 0.048 | 3 | 0.020 | 21 | 2.348 | 0.102 | Used in more than one cluster |
| **SC13** | 0.145 | 3 | 0.059 | 21 | 2.453 | 0.092 | Used in more than one cluster |
| **SC14** | 0.190 | 3 | 0.047 | 21 | 4.019 | 0.021 | Used in one cluster |
| **SC15** | 0.157 | 3 | 0.062 | 21 | 0.539 | 0.084 | Used in more than one cluster |
| **SC16** | 0.050 | 3 | 0.048 | 21 | 1.042 | 0.395 | Used in more than one cluster |
| **SC17** | 0.007 | 3 | 0.025 | 21 | 0.286 | 0.835 | Used in more than one cluster |
| **SC18** | 0.083 | 3 | 0.034 | 21 | 2.452 | 0.092 | Used in more than one cluster |

**Table 13. The number of projects in each cluster.**

| Cluster |  | Based on five criteria | Based on 18 sub-criteria |
|---|---|---|---|
|  | First cluster | 3 | 4 |
|  | Second cluster | 5 | 6 |
|  | Third cluster | 15 | 13 |
|  | Forth cluster | 2 | 2 |
| **Total number of projects** | 25 | | |

**Table 14. The values of the decision matrix.**

| Project | SC1 | SC2 | SC3 | … | SC17 | SC18 |
|---|---|---|---|---|---|---|
| **1** | 6.56 | 5.75 | 5.12 | … | 5.69 | 5.50 |
| **2** | 5.63 | 5.44 | 4.56 | … | 5.87 | 5.19 |
| **3** | 5.94 | 5.31 | 5.19 | … | 5.56 | 4.94 |
| **…** | … | … | … | … | … | … |
| **23** | 5.94 | 5.25 | 4.87 | … | 5.94 | 5.31 |
| **24** | 6.25 | 5.50 | 4.94 | … | 5.69 | 5.25 |
| **25** | 5.75 | 5.44 | 5.06 | … | 5.94 | 5.06 |

## 4.6. Practical implications

The methodology proposed in this research, which is grounded in the principles of clustering and decision-making, is particularly well-suited for the process of selecting a project portfolio. This is especially important in situations where a large number of potential projects are presented to senior managers and decision makers within organizations. By employing

**Table 15. The final ranking of projects using the MULTIMOORA method based on 18 sub-criteria.**

| Project | Ratio system ranking | Reference Point ranking | Full Multiplicative ranking | Final ranking using dominance theory | Final ranking using WASPAS method |
|---|---|---|---|---|---|
| 1 | 11 | 17 | 12 | 12 | 11 |
| 2 | 17 | 25 | 18 | 18 | 17 |
| 3 | 6 | 5 | 4 | 4 | 4 |
| 4 | 16 | 11 | 17 | 16 | 16 |
| 5 | 22 | 20 | 23 | 23 | 21 |
| 6 | 7 | 14 | 9 | 9 | 12 |
| 7 | 21 | 24 | 22 | 22 | 24 |
| 8 | 14 | 11 | 14 | 14 | 14 |
| 9 | 2 | 5 | 2 | 2 | 3 |
| 10 | 19 | 22 | 19 | 19 | 19 |
| 11 | 4 | 10 | 5 | 5 | 5 |
| 12 | 8 | 13 | 7 | 7 | 7 |
| 13 | 3 | 18 | 3 | 3 | 2 |
| 14 | 1 | 14 | 1 | 1 | 1 |
| 15 | 10 | 8 | 10 | 10 | 10 |
| 16 | 18 | 21 | 15 | 17 | 18 |
| 17 | 25 | 5 | 25 | 25 | 25 |
| 18 | 9 | 1 | 8 | 8 | 8 |
| 19 | 23 | 18 | 21 | 21 | 22 |
| 20 | 20 | 9 | 20 | 20 | 20 |
| 21 | 24 | 14 | 24 | 24 | 23 |
| 22 | 12 | 3 | 11 | 11 | 9 |
| 23 | 13 | 4 | 13 | 13 | 13 |
| 24 | 15 | 2 | 16 | 15 | 15 |
| 25 | 5 | 23 | 6 | 6 | 6 |

the K-means clustering method, these decision-makers can effectively identify and group the most similar projects. This grouping helps in organizing the projects into distinct clusters, making the evaluation process more structured and manageable. Once the projects have been categorized into their respective clusters, the next step involves weighing the various selection criteria using the SWARA method. This method provides a systematic approach to assess the importance of different factors influencing the project selection. Ultimately, this comprehensive approach enables decision-makers to easily determine the best cluster of projects to pursue, utilizing the MULTIMOORA method for the final selection. This multi-faceted methodology not only streamlines the decision-making process but also enhances the overall effectiveness of project portfolio management within the organization.

## 5. Conclusion

The objective of the present study was to develop and present a novel methodology for selecting the appropriate project portfolio in project-based organizations. This methodology integrates the K-means clustering algorithm, the SWARA technique, and the MULTIMOORA method to provide a comprehensive and efficient solution for project portfolio selection. While previous studies have focused primarily on individual aspects of project portfolio management, limited attention has been given to the categorization of similar projects. Categorization plays a critical role in improving the efficiency, reliability, and speed of the project selection process, which is one of the key advantages of our approach over other

**Table 16. The ranking of projects in each cluster.**

| Cluster | Project | Ranking |
|---------|---------|---------|
| **First cluster** | A14 | 1 |
| | A1 | 12 |
| | A16 | 17 |
| | A5 | 23 |
| **Second cluster** | A6 | 9 |
| | A8 | 14 |
| | A2 | 18 |
| | A10 | 19 |
| | A7 | 22 |
| | A21 | 24 |
| **Third cluster** | A9 | 2 |
| | A13 | 3 |
| | A3 | 4 |
| | A11 | 5 |
| | A25 | 6 |
| | A12 | 7 |
| | A15 | 10 |
| | A22 | 11 |
| | A23 | 13 |
| | A4 | 16 |
| | A20 | 20 |
| | A19 | 21 |
| | A17 | 25 |
| **Forth cluster** | A18 | 8 |
| | A24 | 15 |

existing methodologies. By grouping similar projects at the outset, decision-makers can streamline the selection process, ultimately saving time and resources. The use of the K-means clustering algorithm in this methodology significantly enhances the efficiency of the process. K-means, known for its speed and scalability, is particularly effective in handling large datasets, making it well-suited for project-based organizations dealing with multiple potential projects. This clustering technique helps organizations evaluate projects quickly by pre-emptively filtering out inefficient or less relevant projects, allowing managers to focus on those that align with strategic goals. The speed of the K-means algorithm is an important factor in project portfolio selection, as it accelerates the decision-making process, which can otherwise be time-consuming in large organizations with numerous projects.

Following the clustering phase, the MULTIMOORA method is employed to rank projects within each cluster based on both qualitative and quantitative criteria. This multi-faceted ranking system enables organizations to evaluate projects from different perspectives, such as cost, strategic fit, and potential impact, simultaneously. By considering these multiple dimensions, the MULTIMOORA method offers a holistic evaluation, allowing for more informed decision-making in selecting projects that align with the organization's objectives. To support the evaluation and weighting of the criteria, the SWARA method is utilized due to its simplicity, speed, and effectiveness. The SWARA technique provides an intuitive, step-by-step approach to determining the relative importance of each criterion, making it a practical choice for fast-paced decision-making environments. The rapid assessment of criteria weights increases the overall efficiency of the project evaluation process, allowing organizations to prioritize projects more quickly and accurately.

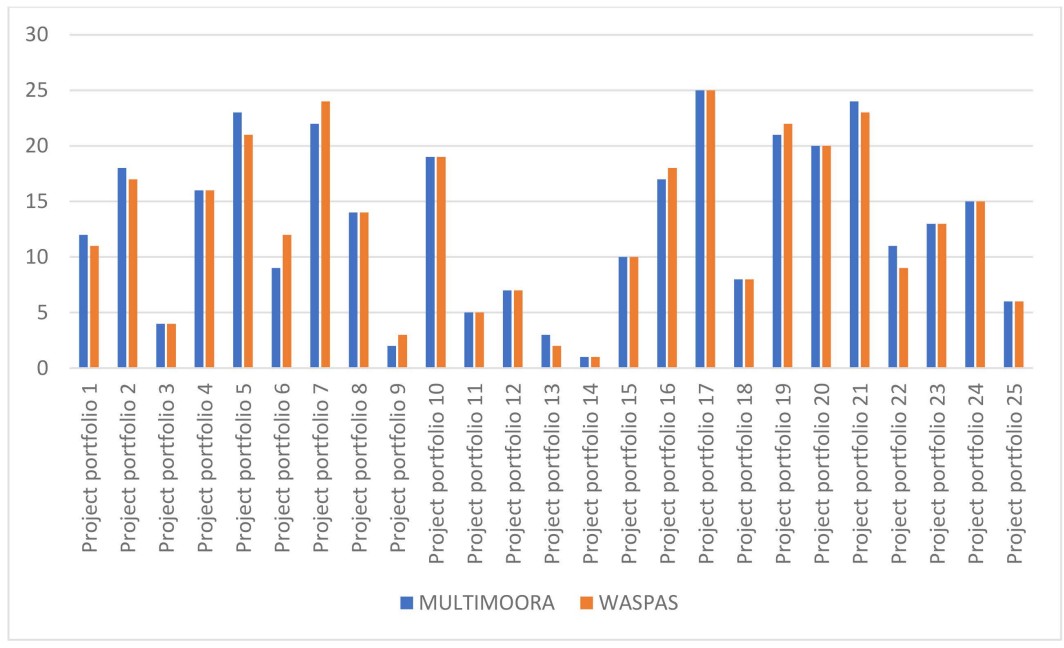

**Fig 3. Comparison of results of MULTIMOORA and WASPAS.**

In this study, the proposed methodology was applied to evaluate 25 projects based on 18 sub-criteria. The projects were then classified into four distinct clusters, and the results were statistically validated using an ANOVA test. Following the clustering phase, the projects within each cluster were ranked based on their relative performance. This approach gives managers the flexibility to choose the highest-scoring projects from each cluster or to select entire clusters that align with the organization's resources and strategic goals. This enables organizations to prioritize and implement projects that best suit their operational capacity and long-term vision.

This study advances project portfolio selection theory by integrating clustering (K-means) with MCDM (SWARA-MULTIMOORA), addressing a critical gap in handling heterogeneous projects. The proposed framework extends prior MCDM models by demonstrating how clustering can enhance decision efficiency without sacrificing robustness, a theoretical bridge between strategic grouping and multi-criteria evaluation.

Despite the strengths of this research, it is important to acknowledge several limitations. One key challenge was the incomplete response from some experts who participated in the study, as well as instances where certain questionnaires were left unfinished. This lack of full participation may have affected the robustness of the data collected. Additionally, the availability of some key managers and experts was constrained, limiting their involvement in the research process. Time constraints also played a role in preventing these stakeholders from fully engaging with the practical applications and proposals derived from the study. As a result, the ability to obtain timely feedback regarding the implementation of the findings within the organization was hindered.

Given these limitations, future research should focus on refining and expanding the methodologies used in this study. Further testing of the proposed approach across different industries would help validate its generalizability and identify sector-specific adjustments that could enhance its applicability. Moreover, integrating additional complementary techniques, such as fuzzy logic or the Analytic Hierarchy Process (AHP), could further enrich the decision-making process by handling uncertainty and subjective assessments more effectively [59,60,61,62,63,64,65,66,67,68,69,70,71,72,73,74,75,76,77,78,79].

**Table 17. The results of the sensitivity analysis.**

| Ranking Project | Final ranking of projects considering all criteria | Ranking by removing financial criterion | Ranking by eliminating technical and practical criterion | Ranking by removing service quality criterion | Ranking by eliminating organizational criterion | Ranking by removing risk criterion |
|---|---|---|---|---|---|---|
| 1 | 12 | 12 | 12 | 12 | 12 | 14 |
| 2 | 18 | 18 | 18 | 18 | 17 | 18 |
| 3 | 4 | 3 | 4 | 3 | 2 | 4 |
| 4 | 16 | 15 | 19 | 16 | 16 | 16 |
| 5 | 23 | 23 | 23 | 23 | 23 | 23 |
| 6 | 9 | 9 | 16 | 9 | 9 | 8 |
| 7 | 22 | 22 | 22 | 22 | 22 | 22 |
| 8 | 14 | 14 | 14 | 14 | 14 | 12 |
| 9 | 2 | 2 | 1 | 2 | 1 | 5 |
| 10 | 19 | 19 | 15 | 19 | 19 | 19 |
| 11 | 5 | 5 | 5 | 5 | 5 | 3 |
| 12 | 7 | 7 | 7 | 7 | 7 | 7 |
| 13 | 3 | 4 | 2 | 4 | 4 | 1 |
| 14 | 1 | 1 | 3 | 1 | 6 | 2 |
| 15 | 10 | 13 | 10 | 10 | 10 | 10 |
| 16 | 17 | 17 | 17 | 17 | 18 | 17 |
| 17 | 25 | 25 | 25 | 25 | 25 | 25 |
| 18 | 8 | 8 | 8 | 6 | 8 | 9 |
| 19 | 21 | 21 | 24 | 21 | 21 | 21 |
| 20 | 20 | 20 | 20 | 20 | 20 | 20 |
| 21 | 24 | 24 | 21 | 24 | 24 | 24 |
| 22 | 11 | 11 | 11 | 11 | 13 | 11 |
| 23 | 13 | 10 | 13 | 13 | 11 | 13 |
| 24 | 15 | 16 | 9 | 15 | 15 | 15 |
| 25 | 6 | 6 | 6 | 8 | 3 | 6 |
| Similarity percentage | | 76% | 64% | 84% | 64% | 68% |

Furthermore, future studies could explore the initial evaluation of projects using mathematical methods like Data Envelopment Analysis (DEA), which could provide a more comprehensive performance assessment of each project. In addition, other clustering methods such as hierarchical clustering and density clustering can be utilized and compared with the K-Means clustering method. Also, optimization models tailored to the goals and constraints of the organization should be developed, enabling the selection of projects based on both strategic objectives and available resources. These advancements in Multi-Criteria Decision-Making (MCDM) practices will contribute to a more sophisticated, adaptable, and reliable approach to project portfolio selection, enhancing the long-term effectiveness of project-based organizations.

## Author contributions

**Conceptualization:** Mohammad Khalilzadeh, Peyman Taebi.

**Data curation:** Mohammad Khalilzadeh, Peyman Taebi.

**Formal analysis:** Mohammad Khalilzadeh.

**Funding acquisition:** Mohammad Khalilzadeh.

**Investigation:** Mohammad Khalilzadeh.

**Methodology:** Mohammad Khalilzadeh, Ali Heidari.

**Project administration:** Mohammad Khalilzadeh.

**Resources:** Mohammad Khalilzadeh, Peyman Taebi.

**Software:** Mohammad Khalilzadeh, Peyman Taebi.

**Supervision:** Mohammad Khalilzadeh.

**Validation:** Mohammad Khalilzadeh, Peyman Taebi, Ali Heidari.

**Visualization:** Mohammad Khalilzadeh.

**Writing – original draft:** Mohammad Khalilzadeh, Peyman Taebi.

**Writing – review & editing:** Mohammad Khalilzadeh, Ali Heidari.

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
