## [Decision Letter · Decision Letter 0]

17 Dec 2024

Dear Dr. Heidari,

Thank you for submitting your manuscript to PLOS ONE. After careful consideration, we feel that it has merit but does not fully meet PLOS ONE’s publication criteria as it currently stands. Therefore, we invite you to submit a revised version of the manuscript that addresses the points raised during the review process.

We look forward to receiving your revised manuscript.

Kind regards,

Muhammet Gul, Ph.D.

Academic Editor

PLOS ONE

Journal Requirements:

2. You indicated that ethical approval was not necessary for your study. We understand that the framework for ethical oversight requirements for studies of this type may differ depending on the setting and we would appreciate some further clarification regarding your research. Could you please provide further details on why your study is exempt from the need for approval and confirmation from your institutional review board or research ethics committee (e.g., in the form of a letter or email correspondence) that ethics review was not necessary for this study? Please include a copy of the correspondence as an ""Other"" file.

4. In the online submission form, you indicated that the data that support the findings of this study are available from the corresponding author upon reasonable request. Data will be made available on request.

Reviewers' comments:

Reviewer's Responses to Questions

**Comments to the Author**

1. Is the manuscript technically sound, and do the data support the conclusions?

Reviewer #1: Yes

Reviewer #2: Yes

Reviewer #3: No

Reviewer #4: Yes

2. Has the statistical analysis been performed appropriately and rigorously?

Reviewer #1: N/A

Reviewer #2: Yes

Reviewer #3: No

Reviewer #4: Yes

3. Have the authors made all data underlying the findings in their manuscript fully available?

Reviewer #1: Yes

Reviewer #2: Yes

Reviewer #3: No

Reviewer #4: Yes

4. Is the manuscript presented in an intelligible fashion and written in standard English?

Reviewer #1: Yes

Reviewer #2: Yes

Reviewer #3: No

Reviewer #4: Yes

Reviewer #1: Despite the fact that the article is fluid and the subject matter is intriguing, the article contains a number of shortcomings, including the following:

1-) Firstly, the introduction is not long enough. There is a need for a very lengthy introduction to the topic.

2) What are the reasons behind the employment of the SWARA and MULTIMOORA methodologies in the work? How come the writers chose to use these particular methods? Please provide an explanation for this.

3-) What does the paper provide to the existing body of research? The extent of the addition to the body of literature is not entirely clear.

4-) The conclusion of the article is not very convincing. This section should be re-written in detail.

Reviewer #2: The article proposes a new method based on clustering and decision-making for project portfolio selection in project-based companies. The method first uses the K-means method to cluster projects, then the SWARA method to determine the weights of the selected criteria, and finally the MULTIMOORA method to rank and select projects.

However, there are still some areas for improvement in the article, for example:

The article should explain how to handle missing values and outliers, for example, whether they are filled in or deleted.

In the design of the model, it is recommended to consider using other clustering methods such as hierarchical clustering or density clustering for comparison to verify the effectiveness of the K-means method, and compare other methods to evaluate the advantages and disadvantages of the MULTIMOORA method.

When displaying research results, it is recommended to use charts or graphs to make the presentation more intuitive.

Reviewer #3: The literature review is not well done. The authors have not considered much research.

The research gaps are not well explored and extracted.

The manuscript is not acceptable in innovation and contribution.

Reviewer #4: The manuscript is well written, well presented, and well organized. The ideas are clear. There are just a few comments

1. After equation (11), "x is the decision matrix" X is a capital letter not small to indicate a matrix.

2. on page 21, ANOVA is written ANNOVA, please correct.

3. Table 15, how did you choose the best alternative whenever the dominance principle is not applicable? For example projects 1, 2, 3, 4, 5, 6, 7, 9 etc. mainly most of the projects.

**Do you want your identity to be public for this peer review?** For information about this choice, including consent withdrawal, please see our Privacy Policy

Reviewer #1: No

Reviewer #2: No

Reviewer #3: No

Reviewer #4: No

---

## [Author Response · Author response to Decision Letter 1]

7 Feb 2025

Authors’ response to the comments

Dear Professor Muhammet Gul

The Academic Editor of “PLOS ONE”,

We appreciate the time and efforts by the Editor and reviewers in reviewing this manuscript. We believe that the manuscript has been greatly improved after the first round of review based on the received comments. We have addressed each of their concerns as outlined below. We are pleased to submit the revised version of the manuscript No. PONE-D-24-48727 entitled: “A novel method based on clustering and decision-making for construction project portfolio selection”. All changes in the revised manuscript are highlighted in YELLOW. We hope that our responses to the comments satisfy your concerns and meet your final approval.

Response to the Comments of Reviewer 1:

Comment 1: Despite the fact that the article is fluid and the subject matter is intriguing, the article contains a number of shortcomings, including the following:

1) Firstly, the introduction is not long enough. There is a need for a very lengthy introduction to the topic.

Response: First of all, thank you very much for taking the time and your constructive and promising comments that helped us improve the paper. The introduction has been revised and lengthen.

Comment 2: What are the reasons behind the employment of the SWARA and MULTIMOORA methodologies in the work? How come the writers chose to use these particular methods? Please provide an explanation for this.

Response: We are very grateful for this insightful comment. The reasons for the employment of SWARA and MULTIMOORA methods have been incorporated into the revised introduction as follows:

First, the K-Means method is applied to categorize projects based on their characteristics, allowing for a clearer understanding of project groupings and strategic relevance. Subsequently, the SWARA and MULTIMOORA methods are utilized to rank and select the most suitable projects for inclusion in the portfolio, based on multiple criteria that align with organizational goals. The SWARA and MULTIMOORA methodologies are employed for project portfolio selection due to their complementary strengths in handling complex, multi-criteria decision-making problems. SWARA is ideal for determining the relative importance of criteria based on expert judgment, making it suitable for projects where subjective assessments, such as strategic alignment and risk, are crucial (Kahraman et al., 2015; Rashidi et al., 2021). MULTIMOORA, on the other hand, excels in evaluating alternatives based on both quantitative and qualitative criteria, offering a holistic approach to ranking and selecting projects (Zavadskas et al., 2016; Sadiq et al., 2020; Chen et al., 2021; Li et al., 2022). Together, these methods provide a comprehensive framework that addresses both the subjective and objective aspects of project portfolio selection, ensuring alignment with organizational goals while effectively managing resources and risks.

By integrating these advanced methods, this research aims to provide a more comprehensive and adaptable framework for project portfolio selection, particularly in environments characterized by resource constraints, uncertainty, and conflicting objectives.

Comment 3: What does the paper provide to the existing body of research? The extent of the addition to the body of literature is not entirely clear.

Response: We are very thankful for your precious comment. The novelties and contributions of the present study have been provided in the introduction and literature review sections.

Comment 4: The conclusion of the article is not very convincing. This section should be re-written in detail.

Response: We highly appreciate this valuable comment. The conclusion has been revised and re-written.

Once again, we would like to express our gratitude for your time and effort dedicated to reviewing our manuscript, as well as the editor's support throughout the process. With the unanimous support from all reviewers, we hope that our manuscript will be accepted for publication in this journal, allowing us to contribute to the scientific community and advance knowledge in our field. Thank you once again for your time, effort, and valuable feedback throughout the review process.

Response to the Comments of Reviewer 2:

Comment: The article proposes a new method based on clustering and decision-making for project portfolio selection in project-based companies. The method first uses the K-means method to cluster projects, then the SWARA method to determine the weights of the selected criteria, and finally the MULTIMOORA method to rank and select projects.

However, there are still some areas for improvement in the article, for example:

Response: First of all, thank you very much for taking the time and your constructive and promising comments that helped us improve the paper. We are delighted that our revised manuscript meets your requirements and satisfies your concern. We also appreciate your thorough evaluation and kind words regarding the improvements made. We believe that the incorporation of your suggestions has significantly strengthened our work and better conveyed our research contributions to the scientific community.

Comment 1: The article should explain how to handle missing values and outliers, for example, whether they are filled in or deleted.

Response: We are very grateful for pointing out the importance of addressing missing values and outliers in the dataset. In the current study, we ensured data quality by taking the following steps: Handling missing values: Missing values were minimal in the collected dataset. When encountered, they were addressed using expert input to fill in the missing data based on the context of the specific project. This approach was chosen to maintain the completeness of the data while preserving its integrity. Handling outliers: Outliers were carefully identified and evaluated in consultation with the experts. These steps ensured that the dataset was robust for subsequent analysis using the K-means clustering together with the MULTIMOORA and WASPAS methods.

Comment 2: In the design of the model, it is recommended to consider using other clustering methods such as hierarchical clustering or density clustering for comparison to verify the effectiveness of the K-means method, and compare other methods to evaluate the advantages and disadvantages of the MULTIMOORA method.

Response: We are very thankful for these valuable suggestions to consider alternative clustering methods, such as hierarchical clustering and density-based clustering as well as the comparison of other methods to evaluate the advantages and disadvantages of the MULTIMOORA method. We fully acknowledge the potential benefits of such an analysis in evaluating the robustness of the proposed methodology. However, the focus of this study is specifically on demonstrating the integration of the K-means clustering algorithm with the SWARA and MULTIMOORA decision-making methods for construction project portfolio selection. K-means was chosen due to its proven efficiency in handling large datasets, its suitability for our specific data characteristics, and its ability to align well with the objectives of the study. While hierarchical and density-based clustering methods are indeed valuable, they differ fundamentally in their assumptions and computational approaches, which may not align with the practical needs and computational simplicity required in our case study. In addition, K-Means clustering offers several advantages over Hierarchical and Density-Based clustering methods, particularly in terms of efficiency, scalability, and simplicity: 1. Efficiency and Scalability: K-Means is computationally efficient and scales well with large datasets. Its time complexity is linear, making it suitable for big data applications. In contrast, Hierarchical clustering has higher computational complexity, often making it impractical for large datasets. 2. Simplicity and Ease of Implementation: The K-Means algorithm is straightforward to understand and implement. Its simplicity allows for easy adaptation and integration into various applications. On the other hand, Density-Based clustering methods can be more complex to implement and may require careful parameter tuning. 3. Interpretability: K-Means provides clear and interpretable results by representing each cluster through its centroid, facilitating easy understanding of the clustering outcome. Hierarchical clustering results in dendrograms, which, while informative, can be more challenging to interpret, especially with large datasets. 4. Performance with High-Dimensional Data: K-Means performs well with high-dimensional data, maintaining efficiency and effectiveness. In contrast, Density-Based clustering methods may struggle with high-dimensional spaces due to the curse of dimensionality, which can affect their performance. In summary, K-Means clustering is advantageous when dealing with large, high-dimensional datasets requiring efficient and straightforward clustering solutions. However, it's essential to consider the nature of the data and the specific requirements of the analysis, as other methods like Hierarchical and Density-Based clustering may be more suitable for datasets with complex, non-spherical cluster shapes or varying densities. Moreover, there are several published papers in the Q1-ranked journals that used only the K-Means clustering method such as the following studies:

de Souto, M. C. P., Costa, I. G., de Araujo, D. S. A., Ludermir, T. B., & Schliep, A. (2008). Clustering cancer gene expression data: A comparative study. BMC Bioinformatics, 9(1), 497. https://doi.org/10.1186/1471-2105-9-497

Godichon-Baggioni, A., Maugis-Rabusseau, C., & Rau, A. (2018). Clustering transformed compositional data using K-means, with applications in gene expression and bicycle sharing system data. Journal of Applied Statistics, 45(12), 2311–2331. https://doi.org/10.1080/02664763.2018.1441383.

Furthermore, the primary aim of this research is to introduce a novel framework rather than conducting an extensive comparative analysis of clustering techniques. To ensure methodological rigor, we have carefully validated the K-means clustering results using expert judgment and statistical techniques (e.g., ANOVA) to confirm the significance and reliability of the clusters formed.

In addition, in order to address this comment, we compared the results of the MUTIMOORA technique and WASPAS ( as a new and efficient MCDM method). Finally, we added your suggestion to the suggestions for further research.

Thank you once again for your thoughtful feedback, which has helped refine the clarity and focus of this research.

Comment 3: When displaying research results, it is recommended to use charts or graphs to make the presentation more intuitive.

Response: We highly appreciate your valuable feedback regarding the recommendation to use charts or graphs to present research results more intuitively. We appreciate your suggestion and fully understand the importance of clear and intuitive data presentation. However, we believe that the use of tables is more suitable for our study for several reasons such as follows: Detailed Information: Tables present precise numerical data, making it easy to observe exact differences and trends that might be lost in graphs or charts. Clarity: With multiple criteria and sub-criteria to compare, tables offer a clear and structured view, avoiding the oversimplification of graphical formats. Industry Standard: Tables are widely accepted in similar studies, facilitating replication and deeper analysis. Space Efficiency: Including both tables and charts would be redundant and exceed manuscript word limits, as tables already provide a complete and concise representation. We trust that the rationale provided above justifies our choice of using tables for data presentation in this study. Nonetheless, we remain open to further suggestions to enhance the clarity of our work.

Once again, we would like to express our gratitude for your time and effort dedicated to reviewing our manuscript, as well as the editor's support throughout the process. With the unanimous support from all reviewers, we hope that our manuscript will be accepted for publication in this journal, allowing us to contribute to the scientific community and advance knowledge in our field. Thank you once again for your time, effort, and valuable feedback throughout the review process.

Response to the Comments of Reviewer 3:

Comment 1: The literature review is not well done. The authors have not considered much research.

Response: First of all, thank you very much for taking the time and your constructive and promising comments that helped us improve the paper. The literature review has been revised and more recent papers have been added and analyzed.

Comment 2: The research gaps are not well explored and extracted.

Response: We are very grateful for your valuable comment. The research gaps have been explored and extracted.

Comment 3: The manuscript is not acceptable in innovation and contribution.

Response: We are very thankful for this insightful comment. The innovations and contributions of the present study have been explained.

Once again, we would like to express our gratitude for your time and effort dedicated to reviewing our manuscript, as well as the editor's support throughout the process. With the unanimous support from all reviewers, we hope that our manuscript will be accepted for publication in this journal, allowing us to contribute to the scientific community and advance knowledge in our field. Thank you once again for your time, effort, and valuable feedback throughout the review process.

Response to the Comments of Reviewer 4:

Comment 1: After equation (11), "x is the decision matrix" X is a capital letter not small to indicate a matrix.

Response: First of all, thank you very much for taking the time and your constructive and promising comments that helped us improve the paper. It has been edited.

Comment 2: on page 21, ANOVA is written ANNOVA, please correct.

Response: We are very grateful for your valuable comment. It has been corrected.

Comment 3: Table 15, how did you choose the best alternative whenever the dominance principle is not applicable? For example, projects 1, 2, 3, 4, 5, 6, 7, 9 etc. mainly most of the projects.

Response: We are very thankful for this insightful comment. If there is no dominance condition, the general dominance condition is used for the ranking of the alternatives. This means that it is sufficient that two out of three ranks of an alternative (assume the alternative A) to be better than the other alternative (assume the alternative B) in order to dominate. When the dominance condition is not met, the final rank is determined as follows: For example, for the alternative (7,7,7), the rank of 7 indicates the dominance. But suppose we have two alternatives as follows: A (7.8.9) and B (5.6.10). In this case, the final rank cannot be determined dominantly and general dominance principle applied because the alternative B is better than the alternative A based on two points 5 and 6 (has a lower rank), so it is the superior alternative )This is mentioned in the step 6 of the MULTIMOORA method.)

Once again, we would like to express our gratitude for your time and effort dedicated to reviewing our manuscript, as well as the editor's support throughout the process. With the unanimous support from all reviewers, we hope that our manuscript will be accepted for publication in this journal, allowing us to contribute to the scientific community and advance knowledge in our field. Thank you once again for your time, effort, and valuable feedback throughout the review process.

---

## [Decision Letter · Decision Letter 1]

25 Feb 2025

Dear Dr. Heidari,

Thank you for submitting your manuscript to PLOS ONE. After careful consideration, we feel that it has merit but does not fully meet PLOS ONE’s publication criteria as it currently stands. Therefore, we invite you to submit a revised version of the manuscript that addresses the points raised during the review process.

We look forward to receiving your revised manuscript.

Kind regards,

Muhammet Gul, Ph.D.

Academic Editor

PLOS ONE

Reviewers' comments:

Reviewer's Responses to Questions

**Comments to the Author**

Reviewer #1: (No Response)

Reviewer #4: All comments have been addressed

2. Is the manuscript technically sound, and do the data support the conclusions?

Reviewer #1: Yes

Reviewer #4: Yes

3. Has the statistical analysis been performed appropriately and rigorously?

Reviewer #1: N/A

Reviewer #4: Yes

4. Have the authors made all data underlying the findings in their manuscript fully available?

Reviewer #1: Yes

Reviewer #4: Yes

5. Is the manuscript presented in an intelligible fashion and written in standard English?

Reviewer #1: Yes

Reviewer #4: Yes

Reviewer #1: The manuscript presents a hybrid decision-making model integrating K-means clustering, SWARA, and MULTIMOORA for construction project portfolio selection. While the study is methodologically rigorous and practically valuable, it requires improvements before publication. The literature review should better highlight its novelty compared to existing MCDM models. The expert selection process needs more transparency, and a sensitivity analysis should be included to assess ranking stability. Additionally, visual representations (graphs, hierarchical trees) would enhance result clarity, and a discussion on theoretical contributions and study limitations should be added. Addressing these major revisions will strengthen the manuscript and enhance its impact in the field of project management and decision-making research.

Reviewer #4: 1. In the added WASPAS, in step 1, what is rij? It is not defined.

2. Similarly, in Step 2, what is tij? it is not defined.

3. "the dominance theory is used to integrate these ratings (Adali & Isik, 2017)."

The dominance principle used in ranking is not present in this reference.

4. It is better to explain the applied dominance principle in step 6 together with the reference .

**Do you want your identity to be public for this peer review?** For information about this choice, including consent withdrawal, please see our Privacy Policy

Reviewer #1: No

Reviewer #4: No

---

## [Author Response · Author response to Decision Letter 2]

9 May 2025

Authors’ response to the comments

Dear Professor Muhammet Gul

The Academic Editor of “PLOS ONE”,

We appreciate the time and efforts by the Editor and reviewers in reviewing this manuscript again. We believe that the manuscript has been greatly improved after the first and second rounds of review based on the received comments. We have addressed each of their concerns as outlined below. We are pleased to submit the second revised version of the manuscript No. PONE-D-24-48727 entitled: “A novel method based on clustering and decision-making for construction project portfolio selection”. All changes in the revised manuscript are highlighted in YELLOW. We hope that our responses to the comments satisfy your concerns and meet your final approval after two rounds of review.

Response to the Comments of Reviewer 1:

General comment: The manuscript presents a hybrid decision-making model integrating K-means clustering, SWARA, and MULTIMOORA for construction project portfolio selection. While the study is methodologically rigorous and practically valuable, it requires improvements before publication.

Response: First of all, thank you very much for taking the time and your constructive and promising comments that helped us improve the paper. We hope to receive your final approval after two rounds.

Comment 1: The literature review should better highlight its novelty compared to existing MCDM models.

Response: We are very grateful for this insightful comment. The novelties and contributions of this research have been highlighted in the literature review.

Comment 2: The expert selection process needs more transparency, and a sensitivity analysis should be included to assess ranking stability.

Response: We are very thankful for your precious comments. The expert selection process has been described and more details have been provided for transparency. In addition, the sensitivity analysis has been included and the ranking stability has been assessed.

Comment 4: Additionally, visual representations (graphs, hierarchical trees) would enhance result clarity, and a discussion on theoretical contributions and study limitations should be added. Addressing these major revisions will strengthen the manuscript and enhance its impact in the field of project management and decision-making research.

Response: We highly appreciate these valuable comments. A comparison chart has been provided for enhancing the clarity of results. In addition, a discussion on theoretical contributions and study limitations has been given in the conclusion section. We should mention that the research limitations and suggestions for future studies have been thoroughly made in the conclusion section.

Once again, we would like to express our gratitude for your time and effort dedicated to reviewing our manuscript, as well as the editor's support throughout the process. With the unanimous support from all reviewers, we hope that our manuscript will be accepted for publication in the PLOS ONE journal, allowing us to contribute to the scientific community and advance knowledge in our field. Thank you once again for your time, effort, and valuable feedback throughout the review process.

Response to the Comments of Reviewer 4:

Comment 1: In the added WASPAS, in step 1, what is rij? It is not defined.

Response: First of all, thank you very much for taking the time and your constructive and promising comments that helped us improve the paper. The definition of rij has been added.

Comment 2: In Step 2, what is tij? it is not defined.

Response: We are very grateful for your valuable comment. tij and the related equation has been edited.

Comment 3: the dominance theory is used to integrate these ratings (Adali & Isik, 2017). The dominance principle used in ranking is not present in this reference.

Response: We are very thankful for this insightful comment. The previous reference (Adali & Isik, 2017) has been replaced with the new references: (Brauers & Zavadskas, 2012; Hafezalkotob et al., 2019).

Comment 4: It is better to explain the applied dominance principle in step 6 together with the reference.

Response: We highly appreciate your precise comment. The explanation has been provided in the second revision.

Once again, we would like to express our gratitude for your time and effort dedicated to reviewing our manuscript, as well as the editor's support throughout the process. With the unanimous support from all reviewers, we hope that our manuscript will be accepted for publication in the PLOS ONE journal, allowing us to contribute to the scientific community and advance knowledge in our field. Thank you once again for your time, effort, and valuable feedback throughout the review process.

---

## [Decision Letter · Decision Letter 2]

9 Jul 2025

Dear Dr. Heidari,

Thank you for submitting your manuscript to PLOS ONE. After careful consideration, we feel that it has merit but does not fully meet PLOS ONE’s publication criteria as it currently stands. Therefore, we invite you to submit a revised version of the manuscript that addresses the points raised during the review process.

We look forward to receiving your revised manuscript.

Kind regards,

Muhammet Gul, Ph.D.

Academic Editor

PLOS ONE

Journal Requirements:

Reviewers' comments:

Reviewer's Responses to Questions

**Comments to the Author**

Reviewer #4: All comments have been addressed

2. Is the manuscript technically sound, and do the data support the conclusions?

Reviewer #4: Yes

3. Has the statistical analysis been performed appropriately and rigorously?

Reviewer #4: Yes

4. Have the authors made all data underlying the findings in their manuscript fully available?

Reviewer #4: Yes

5. Is the manuscript presented in an intelligible fashion and written in standard English?

Reviewer #4: Yes

Reviewer #4: Although the authors made the necessary corrections, there is a comment in regard to these corrections.

rij and tij does not denote matrices, they denote elements of matrices.

The correct from of a matrix is R=[rij], and T=[tij], similar to the matrix X in equation (11), please correct.

**Do you want your identity to be public for this peer review?** For information about this choice, including consent withdrawal, please see our Privacy Policy

Reviewer #4: No

---

## [Author Response · Author response to Decision Letter 3]

2 Nov 2025

Authors’ response to the comments

Dear Professor Muhammet Gul

The Academic Editor of “PLOS ONE”,

We appreciate the time and efforts by the Editor and reviewers in reviewing this manuscript again. We believe that the manuscript has been greatly improved after the first, second, and third rounds of review based on the received comments. We have addressed each of their concerns as outlined below. We are pleased to submit the second revised version of the manuscript No. PONE-D-24-48727 entitled: “A novel method based on clustering and decision-making for construction project portfolio selection”. All changes in the third revision are highlighted in BLUE. Also, we should remind that all previous changes in the first and second revisions have been highlighted in YELLOW. We hope that our responses to the comments satisfy your concerns and meet your final approval after three rounds of review.

Response to the PLOS ONE Journal Requirements:

Comment: Please review your reference list to ensure that it is complete and correct. If you have cited papers that have been retracted, please include the rationale for doing so in the manuscript text, or remove these references and replace them with relevant current references. Any changes to the reference list should be mentioned in the rebuttal letter that accompanies your revised manuscript. If you need to cite a retracted article, indicate the article’s retracted status in the References list and also include a citation and full reference for the retraction notice.

Response: We highly appreciate all your kind attention and taking your precious time for reviewing this manuscript. We have checked the entire manuscript to ensure the reference list is complete.

Once again, we would like to express our gratitude for your valuable effort dedicated to reviewing our manuscript, as well as the editor's support throughout the process. With the unanimous support from all reviewers and the editor, we hope that our manuscript will be accepted for publication in the PLOS ONE journal, allowing us to contribute to the scientific community and advance knowledge in our field. Thank you once again for your time, effort, and valuable feedback throughout the review process.

Response to the Comments of Reviewer 1:

No comment

Response: Once again, we would like to express our gratitude for your time and effort dedicated to reviewing our manuscript, as well as the editor's support throughout the process. With the unanimous support from all reviewers, we hope that our manuscript will be accepted for publication in the PLOS ONE journal, allowing us to contribute to the scientific community and advance knowledge in our field. Thank you once again for your time, effort, and valuable feedback throughout the review process.

Response to the Comments of Reviewer 4:

No comment

Response: Once again, we would like to express our gratitude for your time and effort dedicated to reviewing our manuscript, as well as the editor's support throughout the process. With the unanimous support from all reviewers, we hope that our manuscript will be accepted for publication in the PLOS ONE journal, allowing us to contribute to the scientific community and advance knowledge in our field. Thank you once again for your time, effort, and valuable feedback throughout the review process.

---

## [Decision Letter · Decision Letter 3]

27 Nov 2025

A novel method based on clustering and decision-making for construction project portfolio selection

PONE-D-24-48727R3

Dear Dr. Heidari,

We’re pleased to inform you that your manuscript has been judged scientifically suitable for publication and will be formally accepted for publication once it meets all outstanding technical requirements.

Kind regards,

Muhammet Gul, Ph.D.

Academic Editor

PLOS ONE

Additional Editor Comments (optional):

Reviewers' comments:

Reviewer's Responses to Questions

**Comments to the Author**

Reviewer #4: All comments have been addressed

2. Is the manuscript technically sound, and do the data support the conclusions?

Reviewer #4: Yes

3. Has the statistical analysis been performed appropriately and rigorously?

Reviewer #4: N/A

4. Have the authors made all data underlying the findings in their manuscript fully available?

Reviewer #4: Yes

5. Is the manuscript presented in an intelligible fashion and written in standard English?

Reviewer #4: Yes

Reviewer #4: (No Response)

**Do you want your identity to be public for this peer review?** For information about this choice, including consent withdrawal, please see our Privacy Policy

Reviewer #4: No

---

## [Editor Report · Acceptance letter]

PONE-D-24-48727R3

PLOS One

Dear Dr. Heidari,

I'm pleased to inform you that your manuscript has been deemed suitable for publication in PLOS One. Congratulations! Your manuscript is now being handed over to our production team.

Kind regards,

on behalf of

Dr. Muhammet Gul

Academic Editor

PLOS One